# An Investigation of (Non-) Inclusive Growth in Nigeria's Sub-Nationals: Evidence from Elasticity Approach

**Enobong Udoh [1,\*] and Ndem Ayara [2]**

[1]   Economics Department, Benson Idahosa University, Benin City, P.M.B 1100, Nigeria
[2]   Economics Department, University of Calabar, Calabar, P.M.B 1115, Nigeria; ndemayara1@gmail.com
\*   Correspondence: giftsoncaly@gmail.com

**Abstract:** This paper aims to estimate and rank the performance of sub-nationals in terms of their quality of growth using an index of inequality elasticity of poverty. The study puts forward a scenario matrix to hypothesize the four qualities of growth according to its interactions with inequality and poverty. This model is useful for developing countries that lack GDP data at the sub-national level, provided growth (at the national level) has been positive for the period under review. The study found that for Nigeria's sub-nationals, the null hypothesis of non-inclusive growth was rejected for the different poverty measures.

**Keywords:** elasticities of poverty; inequality; quality of growth; index; scenario matrix; Pareto efficient growth

**JEL Classification:** I32; I38

## 1. Introduction

It is well established in the literature that on average, economic growth is associated with reductions in income poverty (Besley and Burgess 2000; Ravallion 2001; Klasen 2003). Analytically, when the average income of the poor increases in pari passu with growth, it is interpreted as an 'income-effect' of growth. However, whether this income effect truly reduces poverty for the poor remains to be seen. Growth can only be termed inclusive when it reduces poverty. Economic growth in developing countries is desirable and necessary, but it is the distribution of that growth that matters for poverty reduction rather than the pursuit of growth for its own sake. Policymakers often promote economic expansion as the panacea for poverty reduction in developing and emerging markets but fail to recognize that growth is a means to an end, rather than an end in itself.

The traditional focus in poverty literature has been on how economic growth leads to poverty reduction, as it increases per capita real income levels thereby increasing the incomes of the poor. This is referred to as the 'trickle-down effect' of growth, which simply implies a vertical flow of income from the rich to the poor at a given rate (Anderson 1964). In this process, the benefits of economic growth are reaped first by the rich, then subsequently by the poor—once the rich start spending their gains. However, as time went by, there has been a shift in focus of the poverty literature away from the 'trickle-down' concept of growth towards the idea of 'pro-poor growth' (Thornton et al. 1978).

Examining Nigeria's data from Table 1 below, it is clear that apart from the little dent to poverty and inequality by GDP recorded in 2004, on the average, there has been some disconnect in Nigeria's GDP in relation to reducing poverty and closing inequality gaps among the citizenry. Data show that, relative poverty incidence increased even when GDP increased the most in 2010.

**Table 1.** Nigeria's income, inequality, population and GDP distribution profile, 1985–2010.

| Year | Estimated Population (Million) | Gini Index (%) | Relative Poverty Incidence (%) | GDP (Trillion Naira) |
|------|-------------------------------|----------------|-------------------------------|----------------------|
| 1985 | 75 | 38.7 | 46.3 | 0.0679 |
| 1992 | 91.5 | 45.0 | 42.7 | 0.5326 |
| 1996 | 102.3 | 51.9 | 65.6 | 2.7027 |
| 2004 | 126.3 | 42.9 | 54.4 | 11.4111 |
| 2010 | 163 | 44.7 | 69.0 | 33.9847 |

Source: World Bank data (World Development Indicators[1]), NBS Nigeria Poverty Profile, 2012[2]).

Inclusive growth, while requiring poverty reduction, is a broader concept that also focuses on reducing inequalities and different forms of discrimination, including widespread exclusion and unequal access to growth's benefits for women and girls, persons with disabilities, ethnic/linguistic minority groups and entire regions and countries. Inclusive growth also requires full respect for human rights.

As the economies of the world increasingly become globally interconnected, increased poverty and inequality in the developed world can instigate global financial crisis and its resultant tailspin in global growth and employment. In particular, studies have argued that prolonged period of higher inequality in advanced economies was associated with the 2008 global financial crisis (which spread across the globe) by intensifying leverage, overextension of credit, and a relaxation in mortgage-underwriting standards (Rajan 2010). This is very revealing, as it goes to show the debilitating effects of poverty and inequality in the developed world and how it propagates to cause problems for developing countries.

Extreme inequality can instigate conflicts, thereby discouraging investments. Conflicts are particularly prevalent in the management of common resources whereby inequality makes resolving disputes more difficult. For example, the unending militancy in Nigeria's Niger Delta region and Boko Haram insurgency in the north mirrors the deep wounds of extreme poverty and inequality. In a situation where young and energetic boys suffer social and economic exclusion, they have nothing to lose than unleash terror on the nation.

All over the world, government functionaries often chorus that rising GDP is like a tide that will lift all boats (citizens) ashore (prosperity). Evidence from Nigeria however has proved otherwise (Bakare 2014). To this end, it becomes pertinent to investigate further if Nigeria has indeed suffered from non-inclusive growth. Thus, how can Nigeria achieve a faster rate of poverty reduction through policies that will benefit the poorest of the poor? Consequently, what is more interesting in the case of Nigeria is to explore the cross-sub-national variations in its quality of growth—to investigate how pro-poor Nigeria's growth is, looking at poverty not just through the transmission channel of growth but also through inequality.

This study is particularly significant because analysis of (non-) inclusiveness as seen in successful economies are the basis for various cash transfers from the federal government to States. Furthermore, policymakers are likely to be more interested in the percentage point changes (elasticity) in poverty rather than in absolute percent changes.

The study will investigate the following null hypothesis; $H_0$: Growth in the Nigerian economy is non-inclusive at the sub-national level.

This paper is organized as follows. Section two provides a review of the literature, which is broken-down into theoretical and empirical reviews. Section three sets out the methodological framework, which provides the foundations for the parametric analysis to follow and specifies the sources of data used.

---

[1]  https://data.worldbank.org/country/nigeria?view=chart
[2]  http://www.nigerianstat.gov.ng/pdfuploads/Nigeria%20Poverty%20Profile%202010.pdf

Section four presents and analyzes the data and discusses the results. Finally, section five summarizes the findings, draws conclusions and suggests policy recommendations.

## 2. Literature Review

### 2.1. Introduction

Conditional cash transfers have become popular among developing countries but some have argued that its income redistribution effects might reduce poverty at the expense of long-term growth, which makes poverty reduction at the end unsustainable. This is true especially when income redistribution is not productivity raising. If redistribution is used to reduce poverty, be it transitory or structural, then key policy issues are redistribution from whom, to whom, and by what mechanism? The loss and gain of distributive programs on income groups, and their reaction to these losses and gains will depend on the nature of the program. Land redistribution unaccompanied by rural development expenditure might generate a class of poverty-stricken smallholders. Likewise, most of the land redistribution programs in Latin America, even those that radically changed ownership patterns (as in Peru), proved in practice to be poverty generating rather than poverty reducing (Thiesenhusen 1989).

The most frequently advocated manner to achieve such poverty reduction is through economic growth. Growth has therefore traditionally been considered the main engine for poverty reduction. As reported by the World Bank (World Bank 2002), real per-capita income in the developing world grew at an average rate of 2.3% per annum during the four decades between 1960 and 2000. This is a high growth rate by almost any standard. In order to achieve reduction in poverty, however, income growth has to be equitably distributed.

Thus, the current thinking on how best to achieve poverty reduction, both economic growth and equity have to assume a central place in development strategies. Furthermore, equity is seen not only as of intrinsic importance but also of instrumental importance through its impact on the rate at which economic growth leads into poverty reduction. Whether growth reduces poverty and whether in particular growth can be deemed "pro-poor" depends, however, on the impact of growth on inequality and on how much this impact on inequality feeds into poverty (Araar and Duclos 2007). The imperative of growth for combating poverty should not be misinterpreted to mean 'growth is all that matters'. Growth is a necessary condition for poverty alleviation, no doubt, but inequality also matters and should also be on the agenda (Kanbur and Lustig 1999). Growth and distribution are interconnected in numerous ways, and the effectiveness with which growth translates into poverty reduction depends crucially on initial inequality.

The discourse on inequality often makes a distinction between inequality of outcomes (as measured by income, wealth, or expenditure) and inequality of opportunities — attributed to differences in circumstances beyond the individual's control, such as gender, ethnicity, location of birth, or family background. Inequality of outcomes arises from a combination of differences in opportunities and individual's efforts and talent. At the same time, it is not easy to separate effort from opportunity, especially in an intergenerational context. For instance, parental income resulting from their own effort, determines the opportunity of their children to obtain an education. It is in this spirit that Rawls (1971) argued that the distributions of opportunities and of outcomes are equally important and informative to understand the nature and extent of inequality around the world.

### 2.2. Theoretical Literature

This research will draw from the theoretical work by Kakwani and Pernia (2000) who adopted the pioneering elasticity concept of Alfred Marshal to hypothesize a pro-poor growth index. In their paper, Kakwani and Pernia (2000) explained the concept of pro-poor growth, and argued that it represents a major departure from the "trickle-down" phenomenon. They proposed a new indicator—the pro-poor

growth index—that measures the degree to which growth can be considered to be pro-poor with evidence from the nature of economic growth in three countries, namely, Lao PDR, Thailand and Korea.

Broadly, pro-poor growth can be defined as one that enables the poor to actively participate in and significantly benefit from economic activity. It is a major departure from the trickle-down development concept because it is inclusive. Its outcome should be that no person in society is deprived of the minimum basic capabilities. Promoting pro-poor growth requires a strategy that is deliberately biased in favor of the poor so that the poor benefit proportionally more than the rich. Such an outcome would rapidly reduce the incidence of poverty so that those at the bottom end of the distribution curve of consumption would have the resources to meet their minimum basic needs.

### 2.3. Empirical Literature

Bruno, Ravallion and Squire (Bruno et al. 1998), using data from forty-five countries each with at least four distributional surveys over at least two decades found the effect of growth on inequality to be indeterminate. Results from analysis by Bigsten and Shimeles (2005) show that countries with high growth elasticity of poverty indicated that redistribution policies may be effective tools in dealing with poverty in those countries.

Using National Consumer Survey of 1996 and 2003/2004 Nigeria Living Standards Survey, Adigun, Awoyemi and Omonona (Adigun et al. 2011) found that 1 percent increase in income growth would lead to 0.624 percent reduction in poverty. While 1 percent decrease in inequality would decrease poverty by just 0.34 percent. They concluded with the fact that overall rural income distribution did not improve despite government interventions perhaps indicating that the growth process in Nigeria is actually unequalizing.

Kolawole, Omobitan, and Yaqub (Kolawole et al. 2015) used error correction technique to fit time series data from 1980 to 2012. Their results revealed that GDP growth rate increases inequality, but reduces poverty in the country. Nuruddeen and Ibrahim (2014) in their paper used bound testing approach to cointegration and Granger causality test to determine the relationship between poverty, inequality and economic growth in Nigeria. The result found a unidirectional causal relationship running from real GDP to poverty, which means that an increase in real GDP in Nigeria causes high level of poverty. Their paper also indicated that population growth Granger causes literacy without feedback while unidirectional causality exists between poverty and population.

Empirical findings by Bakare (2014) demonstrated a significant and direct relationship between economic growth and poverty in Nigeria, implying that economic growth does not reduce poverty in Nigeria. In other words, the impressive growth of the economy in recent times could not yield an improvement in poverty. The so-called "trickle-down" phenomenon, underlying the view that growth improves poverty is not supported by Nigeria's data. His findings and conclusions suggested the need for policymakers to ensure equitable distribution and allocation of national income. In a similar case for Nigeria, Kolawole and Omobitan (2014) found a negatively significant effect of growth on poverty. As growth increases, poverty declines in the country.

### 2.4. Gap in Literature

Most past works in this area concentrated on OLS regression in estimating elasticities of poverty. For Nigeria, this portends two major problems. Firstly, data by NBS are from field surveys or cyclical data undertaken every six years, which suffers micronumerosity. Secondly, this data is characterized by structural breaks and regime change, which these regressions rarely adjust for. Conversely, where a deterministic model is used, point elasticity is employed which fails the symmetry rule or renders the functional form indeterminate. In addition, it is important that new knowledge in this area goes deeper into a disaggregated analysis (in sub-nationals) for a better understanding of the growth, poverty and inequality nexus.

## 3. Methodology

### 3.1. Data Sources/Description

The data for this study was obtained from the National Bureau of Statistics (NBS) databank. The data comprises Nigeria Living Standard Survey (NLSS) for 2009/10 and 2003/04. The poverty lines for each of the measures according to NBS are as follows:

i.      Absolute Poverty line is N54, 401.16. Absolute (Objective) poverty measure considers both food expenditure and non-food expenditure using annualized per capita expenditure approach. Households that spend below this are categorized under this measure.

ii.     The Relative Poverty line is N66, 802.20. This line separates the poor from the non-poor. All households whose per capita expenditure is less than this are considered poor while those above the stated amount are considered non-poor.

iii.    The Dollar Per day Poverty line is N54, 750. This measure considers all individuals whose expenditure per day is less than a dollar per day. Using the official exchange rate of Naira to Dollar in 2009/2010 was annualized to this amount.

iv.    Another critical measure of poverty is the Gini Coefficient (inequality measurement). This measure explains the spread of income or expenditure among households.

### 3.2. Theoretical Background of the Model

This paper's approach is based on decomposition of poverty into growth and inequality components (Kraay 2004). The important thing with this approach is that its data requirement is minimal (two period information is sufficient).

Arc Elasticity

In analyzing the relationship between our independent variables (inequality and growth) and the dependent variable poverty, literature has shown that their functions are indeterminate or ambiguous. In that regard, it is imperative that the symmetry property of elasticity must be upheld. In mathematics and economics, the arc elasticity captures the responsiveness of one variable to another between two given points. It is the ratio of the percentage change of one of the variables between the two points to the percentage change of the other variable. Allen (1933) advocated the use of the midpoint or arc elasticity formula, which contrasts with the point elasticity, for use due to the following properties:

i.      It is symmetric with respect to the two variables, and
ii.     it is independent of the units of measurement.

The arc elasticity is used when there is not a general function for the relationship of two variables, but two points in the relationship are known. In contrast, calculation of the point elasticity requires detailed knowledge of the functional relationship and can be calculated wherever the function is defined.

The midpoint method can be used if just two points on the curve are known. A person need not know the functional relationship before employing this method. Whereas, in order to calculate the point elasticity a functional relationship between the variables is required. As the difference between the two dependent or independent variable increases, the accuracy of the formula decreases. This happens because the percentage change on the curve is not symmetric.

### 3.3. Model Specification

Ekanem and Iyoha (2000) gave the arc elasticity formula, which was adapted to our model as;

$$\eta_g = \frac{\frac{\Delta\, P_t}{P_t + P_{t+1}}}{\frac{\Delta G_t}{G_t + G_{t+1}}} \tag{1}$$

$$\eta_i = \frac{\frac{\Delta\, P_t}{P_t + P_{t+1}}}{\frac{\Delta Gini_t}{Gini_t + Gini_{t+1}}} \tag{2}$$

where $\eta_g$ is coefficient of growth elasticity of poverty; $\eta_i$ is coefficient of inequality elasticity of poverty; $P_t$ is the poverty rate; $G_t$ is the real GDP; $Gini_t$ is inequality coefficient, $t$ is year; $0 \leq Gini \leq 1$. Based on the theoretical underpinning of the model, it is expected a priori that when $\Delta G > 0$, $\Delta P < 0$ and $\Delta Gini < 0$, $\Delta P < 0$ (ideal). Furthermore, it is important to add that our elasticity model in Equations (1) and (2) are not static but dynamic models because they build in a time lag of six years. This is true as the variables are not all in levels; this is what is called dynamic elasticities. (See Nymoen (2004), page 9).

### 3.4. Operationalization of the Inequality Elasticity of Poverty Index

The purpose of the index is to find a way to explain the quality of growth in sub-nationals given that there has never been GDP data at that level. Thus, was growth pro-poor or pro-rich? This will be answered based on some assumptions.

The assumptions are;

i.     A positive real GDP (increasing function) for all sub-nationals in 2010. This is true as Nigeria had increased growth for the period under review as can be seen from the data Table in Appendix A.
ii.    Increase in growth or incomes always lead to a reduction in poverty holding inequality constant (Kanbur 2004).
iii.   Notwithstanding, inequality is a fact of life and can never be nonexistent, this is supported by the fact that talent, endowments and motivations always differ.

Moreover, literature has also proved that increase in inequality may have positive effects on poverty (against popular belief) although governments always seek to reduce 'inequality of opportunities' as a policy (through spread of welfare projects) (Forbes 2000; Li and Zou 1998).

A simple poverty model is presented below;

$$Poverty = f(GDP, gini) \tag{3}$$

Economic intuition is supported by empirical evidence, which suggests that higher income inequality is associated with a lower growth elasticity of poverty. For our analysis, since we have assumed growth in income to be increasing, likewise, it becomes the necessary condition and income distribution (or inequality) is the sufficient condition. As shown below, poverty rate is explained by growth elasticity (income effect) and inequality elasticity (distributional effect) (Bourguignon 2004).

$$P \text{ (poverty rate)} = \eta_g + \eta_i \tag{4}$$

Scenarios

In tracking the quality of growth, it is maintained that growth increased for the period under review. The scenarios in Table 2 will guide this study to define the quality of growth for our analysis. The following scenarios could suffice within the review period;

i.     Inequality can increase while poverty reduces. Let us call this Pareto efficient growth (for the poor). Here growth is inclusive.
ii.    Inequality can reduce while poverty can increase. Let us call this Pareto inefficient growth amounting to non-inclusive growth.
iii.   Inequality can increase while poverty increases. We call this Pro-rich or non-inclusive growth.
iv.    Inequality can reduce while growth also reduces. We call this Pro-poor or inclusive growth.

**Table 2.** Scenario matrix.

| Variables | Scenario 1 | Scenario 2 | Scenario 3 | Scenario 4 |
|---|---|---|---|---|
| Δ Inequality | + | - | + | - |
| Δ Poverty | - | + | + | - |
| Δ Growth | + | + | + | + |
| Qualities of growth | Pareto efficient growth | Pareto inefficient growth | Pro-rich growth | Pro-poor growth |

Source: Author's compilation.

From this it can be seen that the order for the most desirable quality of growth is the Pro-poor growth than the Pareto efficient growth respectively. The Pareto inefficient and Pro-rich growths are undesirable in that respective order. Therefore, in ranking sub-nationals in the index, priority will first be given to the most desired quality of growth down to the least undesirable quality of growth. Once the sub-nationals have been grouped in their respective categories, the coefficients of their inequality elasticity of poverty will be used as weights in a decreasing order.

**Table 3.** Results for the Coefficients of Inequality Elasticity of Poverty and Quality of Growth Derivation for Sub-Nationals, 2004/2010.

| States (Alphabetical Order) | Absolute Poverty Measure | | | | Relative Poverty Measure | | | | Dollar/Day Measure | | | |
|---|---|---|---|---|---|---|---|---|---|---|---|
| | Direction of Δ Gini (1) | Absolute Poverty η (2) | Direction of Δ Poverty (3) | Quality of Growth (4) | Relative Poverty η (5) | Direction of Δ Poverty (6) | Quality of Growth (7) | Dollar/day Poverty η (8) | Direction of Δ Poverty (9) | Quality of Growth (10) |
| Abia | + | 1.722 | + | PR | 8.101 | + | PR | 5.858 | + | PR |
| Adamawa | − | −0.907 | + | PI | −6.867 | + | PI | −4.392 | + | PI |
| Akwa Ibom | + | −0.586 | − | PE | 3.126 | + | PR | 0.848 | + | PR |
| Anambra | + | 3.528 | + | PR | 14.82 | + | PR | 8.392 | + | PR |
| Bauchi | − | 0.131 | − | PP | 0.090 | − | PP | 0.135 | − | PP |
| Bayelsa | + | 8.626 | + | PR | 88.21 | + | PR | 51.19 | + | PR |
| Benue | + | 2.829 | + | PR | 6.375 | + | PR | 9.732 | + | PR |
| Borno | + | 0.206 | + | PR | 2.019 | + | PR | 1.928 | + | PR |
| Cross−River | + | −1.102 | − | PE | 3.802 | + | PR | 0.257 | + | PR |
| Delta | + | −1.001 | − | PE | 1.590 | + | PR | 0.078 | + | PR |
| Ebonyi | + | 1.623 | + | PR | 3.606 | + | PR | 2.770 | + | PR |
| Edo | + | 1.624 | + | PR | 6.795 | + | PR | 3.580 | + | PR |
| Ekiti | + | −0.290 | − | PE | 1.246 | + | PR | 1.455 | + | PR |
| Enugu | + | 2.606 | + | PR | 11.03 | + | PR | 8.425 | + | PR |
| Gombe | + | 0.765 | + | PR | 0.248 | + | PR | 0.779 | + | PR |
| Imo | + | −1.690 | − | PE | 7.041 | + | PR | 6.262 | + | PR |
| Jigawa | + | −0.446 | − | PE | −1.115 | − | PE | −1.131 | − | PE |
| Kaduna | + | 1.888 | + | PR | 4.205 | + | PR | 5.509 | + | PR |
| Kano | + | 0.759 | + | PR | 0.739 | + | PR | 1.535 | + | PR |
| Katsina | − | −0.569 | + | PI | −1.303 | + | PI | −1.939 | + | PI |
| Kebbi | + | −3.317 | − | PE | −1.591 | − | PE | −2.555 | − | PE |
| Kogi | − | 1.806 | − | PP | 1.094 | − | PP | 1.535 | − | PP |
| Kwara | − | 0.661 | − | PP | 0.461 | − | PP | 0.847 | − | PP |
| Lagos | − | 1.759 | − | PP | 0.237 | − | PP | 0.863 | − | PP |
| Nassarawa | − | −6.242 | + | PI | −5.562 | + | PI | −8.261 | + | PI |
| Niger | + | −85.23 | − | PE | −138.6 | − | PE | −180.5 | − | PE |
| Ogun | + | 6.275 | + | PR | 32.42 | + | PR | 30.99 | + | PR |
| Ondo | + | −0.508 | − | PE | 1.799 | + | PR | 0.635 | + | PR |
| Osun | + | −1.696 | − | PE | 3.723 | + | PR | 4.986 | + | PR |
| Oyo | + | 1.656 | + | PR | 4.965 | + | PR | 5.258 | + | PR |
| Plateau | − | −0.923 | + | PI | −4.602 | + | PI | −7.664 | + | PI |
| Rivers | + | −1.409 | − | PE | 5.189 | + | PR | 1.231 | + | PR |

| States (Alphabetical Order) | Absolute Poverty Measure | | | | Relative Poverty Measure | | | | Dollar/Day Measure | | | |
|---|---|---|---|---|---|---|---|---|---|---|---|
| | Direction of Δ Gini (1) | Absolute Poverty η (2) | Direction of Δ Poverty (3) | Quality of Growth (4) | Relative Poverty η (5) | Direction of Δ Poverty (6) | Quality of Growth (7) | Dollar/day Poverty η (8) | Direction of Δ Poverty (9) | Quality of Growth (10) |
| Sokoto | − | −20.05 | + | PI | −17.44 | + | PI | −22.12 | + | PI |
| Taraba | + | 0.342 | + | PR | 0.577 | + | PR | 0.681 | + | PR |
| Yobe | + | −0.162 | − | PE | −0.098 | − | PE | −0.0006 | − | PE |
| Zamfara | − | 6.897 | − | PP | 0.287 | − | PP | 0.910 | − | PP |
| FCT | + | −0.687 | − | PE | 1.399 | + | PR | 0.732 | + | PR |

Source: Author's compilation. Note; PR = Pro-rich growth; PP = Pro-poor growth; PI = Pareto inefficient growth; PE = Pareto efficient growth (for the poor).

## 4. Results and Discussion

### 4.1. Parametric Analysis

The results of the coefficient of elasticities for the sub-nationals are presented Table 3.

#### 4.1.1. Explanation of the Columns

In explaining Table 3 above, it is imperative to walk through how the various columns were derived. Column 1 shows the changes in Gini either positive or negative for the different States, comparing data for 2004 and 2010. Column 2 shows the computational result of inequality elasticity of poverty for the absolute poverty benchmark using the arc elasticity formula. It ranges between negative and positive. It will serve as weights in the final ranking of States. Column 3 gives the result for the change in poverty comparing 2004 and 2010. Column 4 shows the quality of growth definitions, as shown in the scenario matrix in the methodology given the interaction between inequality, poverty and growth (assumed to be a positive constant). Column 5 gives similar computations as column 2 but only that it measures relative instead of absolute poverty. Column 6 shows the changes in relative poverty between 2004 and 2010. Column 7 similar to column 4 gives the definitions of the interactions between our variables. Column 8 is similar to column 2 and 5 but uses a different poverty measure which is the dollar/day. Column 9 shows the directional change in dollar/day from 2004 and 2010. Column 10 defines the quality of growth using the dollar/day benchmark.

#### 4.1.2. Explanation of the Interaction of the Variables

In discussing the interaction of variables in Table 3 and how the different qualities of growth were derived, it is only proper to align the explanations according to the different types of poverty. First, we start with absolute poverty.

#### 4.1.3. Absolute Poverty Measure

According to results from Table 3, Abia State (known for trade/commerce in the Eastern region) for instance had an increase in its Gini index as seen in column 1. Thus, a 1% increase in inequality resulted in a poverty elasticity of 1.7, overall increasing poverty in marginal terms as shown in column 3. Consequently, combining our earlier assumption of a constant positive growth function with a positive Gini (in column 1) and positive change in poverty (in column 3) results in a pro-rich quality of growth (non-inclusive) which is undesirable.

Likewise, Adamawa (an agrarian State in the Northeast) had a negative change in Gini for the review period (column 1). Thus, a 1% decrease in inequality resulted in a poverty elasticity of −0.9, which was not a sufficient condition to reduce poverty. In fact, poverty even increased during the review period (column 3). Hence, Adamawa's quality of growth as shown in column 4 is Pareto inefficient because though inequality reduced, more citizens were plunged into poverty.

Akwa Ibom State is next. The oil/gas rich State had an increase in Gini whereby a 1% increase in Gini resulted in a poverty elasticity of −0.58, which was sufficient to reduce poverty (column 3). Therefore, its quality of growth was Pareto efficient because though inequality increased in the presence of increased growth, poverty reduced.

Our last State under this benchmark is Bauchi State (an agrarian State), which had a pro-poor (inclusive growth). This was because a 1% reduction in Gini gave 0.13 poverty elasticity, which was not enough to stop the marginal reduction in poverty (column 3). Thus, when growth increased, inequality and poverty reduced giving a desirable growth condition of inclusiveness.

#### 4.1.4. Relative Poverty Measure

Abia State posted an increase in its Gini index (column 1). Thus, a 1% increase in inequality resulted in a poverty elasticity of 8.1, overall increasing poverty in marginal terms (column 6).

Consequently, given a constant positive growth function combined with an increase in Gini (column 1) and positive change in poverty (in column 6) results in a pro-rich quality of growth (non-inclusive) which is not desirable.

Adamawa State had a negative change in Gini for the review period (column 1). Thus, a 1% decrease in inequality resulted in a poverty elasticity of −6.9, which was not a sufficient condition to reduce poverty as poverty even increased for the review period (column 6). Hence, Adamawa's quality of growth as shown in column 7 is Pareto inefficient because though inequality reduced, more citizens were plunged into poverty.

Bauchi State comes next as it presents a different quality of growth from the previous two above. The State had a decrease in Gini whereby a 1% decrease in Gini resulted in a poverty elasticity of 0.09, all the same insufficient to impede poverty reduction (column 6). As evidence shows, poverty reduced in the period under review, as the quality of growth was pro-poor.

Our last State under this benchmark is Jigawa (an agrarian State), which had a Pareto efficient growth. This was because a 1% increase in Gini gave −1.1 in poverty elasticity, which was enough to engender marginal reduction in poverty (column 7). Thus, when growth increased, inequality increased while poverty reduced.

### 4.1.5. Dollar/Day Poverty Measure

Abia State had an increase in its Gini index (column 1). Thus, a 1% increase in inequality resulted in a poverty elasticity of 5.8, overall, increasing poverty in marginal terms (column 9). Consequently, given a constant positive growth function, a positive Gini (column 1) and a positive change in poverty (in column 9) results in a pro-rich quality of growth (non-inclusive) which is not desirable.

Next, Adamawa State had a negative change in Gini for the review period (column 1). Thus, a 1% decrease in inequality resulted in a poverty elasticity of −4.4, which was not sufficient to reduce poverty as poverty even increased for the review period (column 9). Hence, Adamawa's quality of growth as shown in column 10 is Pareto inefficient because though inequality reduced, incidence of poverty escalated.

Bauchi State comes next as it presents a different quality of growth from the previous two above. A 1% decrease in Gini resulted in a poverty elasticity of 0.14, all the same insufficient to affect poverty reduction (column 9). As evidence shows, poverty reduced in the period under review, as the quality of growth was pro-poor.

Lastly, Jigawa State had a Pareto efficient growth. This was because a 1% increase in Gini gave –1.1 in poverty elasticity, which was enough to engender marginal reduction in poverty (column 9). Thus, when growth increased, inequality increased while poverty reduced.

After discussing Table 3, its results are used to calibrate an index for sub-nationals in order to answer the research hypothesis. As presented in Table 4, States are ranked first by their quality of growth and then by the size of their elasticities.

**Table 4.** Ranking of States Performance on Poverty Reduction Based on Different Measures, 2004/2010.

| Rank | Absolute Poverty | Quality of Growth | Relative Poverty | Quality of Growth | Dollar/Day Poverty | Quality of Growth |
|------|------------------|-------------------|------------------|-------------------|--------------------|-------------------|
| 1 | Zamfara | Pro-poor | Kogi | Pro-poor | Kogi | Pro-poor |
| 2 | Kogi | Pro-poor | Kwara | Pro-poor | Zamfara | Pro-poor |
| 3 | Lagos | Pro-poor | Zamfara | Pro-poor | Lagos | Pro-poor |
| 4 | Kwara | Pro-poor | Lagos | Pro-poor | Kwara | Pro-poor |
| 5 | Niger | Pareto Efficient | Bauchi | Pro-poor | Bauchi | Pro-poor |
| 6 | Kebbi | Pareto Efficient | Niger | Pareto Efficient | Niger | Pareto Efficient |
| 7 | Osun | Pareto Efficient | Kebbi | Pareto Efficient | Kebbi | Pareto Efficient |
| 8 | Imo | Pareto Efficient | Jigawa | Pareto Efficient | Jigawa | Pareto Efficient |
| 9 | Rivers | Pareto Efficient | Yobe | Pareto Efficient | Yobe | Pareto Efficient |
| 10 | Cross River | Pareto Efficient | Sokoto | Pareto Inefficient | Sokoto | Pareto Inefficient |
| 11 | Delta | Pareto Efficient | Adamawa | Pareto Inefficient | Nassarawa | Pareto Inefficient |
| 12 | FCT | Pareto Efficient | Nassarawa | Pareto Inefficient | Plateau | Pareto Inefficient |
| 13 | Akwa Ibom | Pareto Efficient | Plateau | Pareto Inefficient | Adamawa | Pareto Inefficient |
| 14 | Ondo | Pareto Efficient | Katsina | Pareto Inefficient | Katsina | Pareto Inefficient |
| 15 | Jigawa | Pareto Efficient | Bayelsa | Pro-rich | Bayelsa | Pro-rich |

**Table 4.** *Cont.*

| Rank | Absolute Poverty | Quality of Growth | Relative Poverty | Quality of Growth | Dollar/Day Poverty | Quality of Growth |
|------|------------------|-------------------|------------------|-------------------|--------------------|-------------------|
| 16 | Ekiti | Pareto Efficient | Ogun | Pro-rich | Benue | Pro-rich |
| 17 | Yobe | Pareto Efficient | Anambra | Pro-rich | Enugu | Pro-rich |
| 18 | Sokoto | Pareto Inefficient | Enugu | Pro-rich | Anambra | Pro-rich |
| 19 | Nassarawa | Pareto Inefficient | Abia | Pro-rich | Imo | Pro-rich |
| 20 | Plateau | Pareto Inefficient | Imo | Pro-rich | Abia | Pro-rich |
| 21 | Adamawa | Pareto Inefficient | Edo | Pro-rich | Kaduna | Pro-rich |
| 22 | Katsina | Pareto Inefficient | Benue | Pro-rich | Oyo | Pro-rich |
| 23 | Bayelsa | Pro-rich | Rivers | Pro-rich | Osun | Pro-rich |
| 24 | Ogun | Pro-rich | Oyo | Pro-rich | Edo | Pro-rich |
| 25 | Anambra | Pro-rich | Kaduna | Pro-rich | Ebonyi | Pro-rich |
| 26 | Benue | Pro-rich | Cross River | Pro-rich | Borno | Pro-rich |
| 27 | Enugu | Pro-rich | Osun | Pro-rich | Kano | Pro-rich |
| 28 | Kaduna | Pro-rich | Ebonyi | Pro-rich | Ekiti | Pro-rich |
| 29 | Abia | Pro-rich | Akwa Ibom | Pro-rich | Rivers | Pro-rich |
| 30 | Oyo | Pro-rich | Borno | Pro-rich | Akwa Ibom | Pro-rich |
| 31 | Edo | Pro-rich | Ondo | Pro-rich | Gombe | Pro-rich |
| 32 | Ebonyi | Pro-rich | Kano | Pro-rich | FCT | Pro-rich |
| 33 | Gombe | Pro-rich | Delta | Pro-rich | Taraba | Pro-rich |
| 34 | Kano | Pro-rich | FCT | Pro-rich | Ondo | Pro-rich |
| 35 | Taraba | Pro-rich | Ekiti | Pro-rich | Cross River | Pro-rich |
| 36 | Borno | Pro-rich | Taraba | Pro-rich | Delta | Pro-rich |

Source: Author's compilation.

Depending on the particular measure of poverty, Table 4 shows the ranking of States, first according to their quality of growth and then by the size of their elasticity coefficient. As shown, the States that have the most desirable quality of growth (inclusiveness) with the highest inequality elasticity of poverty comprise; Zamfara, Kogi, Lagos and Kwara respectively for the absolute poverty measure. While Kogi, Kwara, Zamfara, Lagos and Bauchi respectively for the relative poverty measure. Lastly, Kogi, Zamfara, Lagos, Kwara and Bauchi respectively for the dollar/day measure. The States that suffered the most non-inclusive growth comprise; Borno, Taraba and Kano States respectively for the absolute poverty measure. Then, Taraba, Ekiti and FCT respectively for the relative poverty measure. Lastly, Delta, Cross River and Ondo respectively for the dollar/day poverty measure. A detail poverty mapping for the five best and worst performers according to the different poverty measures is presented in Appendix B as Figures A1–A3.

It can be seen that States like Kogi, Zamfara (both agrarian States) and Lagos (commercial capital of Nigeria) happen to be best performers in all measures of poverty. To explain this, in the case of Kogi and Zamfara for instance, the income distribution among households is not too severe or far apart from its mean income. A reason for this possibility is because of their communal way of life like self-help type of community projects among the rural people. For Lagos, one can notice how Lagos made the greatest leap from 25th position in 2004 to rank 3rd in absolute measure of poverty from NBS 2010 survey results. A key reason for this is how Lagos has fostered itself as the commercial hub of the country. Regrettably, Taraba State was recurring as a worst performer in all the poverty measures, which should be troubling for policymakers.

This index portends great insight for policymakers on how to solve the problem of non-inclusive growth. The analysis has shown that inequality should be seen as a serious problem than the mere overconcentration on growth for which varying type of policies must be implemented to reverse the trend. The options available to policymakers are twofold. First, since pro-rich and Pareto inefficient growth exacerbate poverty, they are not desired, Sates in those categories must be moved into the Pareto efficient (for the poor) group at least or into pro-poor group at best. Secondly, the type of policy choices to be undertaken will be different. For example, the type of policies needed in moving Borno or Taraba States from undesirable type of growth to the desirable type will differ from that needed in an already pro-poor growth State. This be shall highlighted more in the next section.

*4.2. Test of Hypothesis*

Table 5 is a summary classification of Table 4 whereby sub-nationals were grouped into inclusive and non-inclusive growth. Pro-poor and Pareto efficient quality of growth were seen to be desirable while Pareto inefficient and pro-rich quality of growth were undesirable. From the empirical evidence, only 54% (required outcome of 20 States out of 37 total outcomes) suffered non-inclusive growth using the absolute poverty benchmark, 76% (28 States out of 37) suffered non-inclusiveness using the relative poverty benchmark and 76% (28 States out of 37) suffered non-inclusiveness using the dollar/day poverty benchmark. It is clear from the results that non-inclusive growth is not characterized in all sub-nationals so we can confidently reject the null hypothesis, which says sub-nationals are characterized by non-inclusive growth.

**Table 5.** Summary of Number of Sub-nationals with Desirable/Undesirable Quality of Growth.

| Quality of Growth | Absolute Poverty | Relative Poverty | Dollar/Day Poverty |
|---|---|---|---|
| Pro-poor | 4 | 5 | 5 |
| Pareto Efficient | 13 | 4 | 4 |
| Total | 17 | 9 | 9 |
| % inclusive growth | 45.9% | 24.3% | 24.3% |
| % non-inclusive growth | 54.1% | 75.7% | 75.7% |

Source: Author's compilation.

It is important to state that the significance of the sub-national measure over the popular national measure is that the aggregation (or mean value) of sub-nationals data gives the national data. Therefore, analysis of sub-nationals gives a deeper reflection of the quality of growth characterized in Nigeria. As it is clear that the national data loses some strength because it uses mean values.

Going forward, these results portend serious implications for policy. Since different interventions will be needed by respective sub-national governments, it is apt to conclude this section with implications for policy depending on the type of quality of growth that prevails as presented in Table 6.

**Table 6.** Implications for Policy.

| Quality of Growth | Policy Advice |
|---|---|
| Pro-poor growth | Priority here should be to increase growth while minimizing inequality i.e., keep status quo. |
| Pareto efficient growth (for the poor) | The goal here should fully be on reducing inequality without which increased growth will not fully impact on poverty. |
| Pareto inefficient growth (for the poor) | The focus here should be to look for entirely new sources of growth that is different from previous. E.g., transiting from capital to labour intensive production or vice versa. |
| Pro-rich growth | The key priority here should be a simultaneous wealth redistribution/inequality reducing policies. |

## 5. Conclusions and Recommendations

*5.1. Conclusions*

In conclusion, this study found that only absolute poverty had a Pareto-efficient growth while relative and dollar/day measures of poverty all recorded non-inclusive growth from the national perspective. The sub-nationals had evidence of inclusive growth in at most 17 States. Using the quality of growth hypothesis propounded in this study, the results show that Kakwani and Pernia (2000) pro-poor growth index methodology is inadequate in defining pro-poor growth solely based on the size of elasticity coefficients. In contribution to knowledge, this study recognizes that the nexus between

poverty, growth and inequality gives indeterminate results therefore the use of point elasticity like in other studies will give misleading results. The study uses arc elasticity, which recognizes the symmetry rule. Furthermore, the study shows how (non-)inclusive growth can further be disaggregated to the sub-nationals whereas other studies concentrated on aggregate national data.

This study concludes that inequality also matters in defining the quality of growth, as growth is the necessary condition while inequality is the sufficient condition and suggests further detailed analysis with longitudinal survey in the States that were characterized by inclusive growth in order to better explain how their success stories were achieved to encourage diffusion of best practices.

### 5.2. Recommendations

Since the majority of Nigeria's sub-nationals fall into the pro-rich growth category, greater wealth redistribution reform is imperative. Furthermore, since land ownership is a basic factor of production, reform in land ownership will greatly reduce inequality. After all, land reforms in China, Korea and Japan (though radical) helped generate more poverty reduction and subsequently helped generate more pro-poor growth. Education and literacy reduces inequality of opportunities because it increases the pool of people who can access better employment and creates a larger pool of potential entrepreneurs who can set up business that use modern technology thereby fast tracking growth. The government will do well in increasing the budget of this sector.

Although few States like Ekiti, Osun and Cross River have enacted conditional cash transfer schemes, more States will do well to provide social safety nets in order to counter the adverse effects of massive unemployment and galloping inflation (inflation > 10% per year) which contribute to exacerbate poverty and inequality. Of course, there are other regulations in the labor sphere, like local content policy, which can boost internalization of knowledge and skills. Pro-poor growth is not an accidental byproduct of the growth process—conscious policies can help create it.

Lastly, adequate and timely data is beneficial in shaping public policy that will solve the poverty debacle. The government must make funding of NBS a statutory first line charge so that policy can be guided by up to date poverty and inequality data to forestall policymakers groping in the dark.

**Acknowledgments:** The authors appreciate the meticulousness of the reviewers and editor. Any other omissions are that of the authors.

**Author Contributions:** The authors equally contributed to the paper.

**Conflicts of Interest:** The authors declare no conflict of interest.

**Appendix A.**

**Table A1.** Income inequality for 2004/2010.

| Entities | Income Inequality | | % Change from 2004–2010 |
|---|---|---|---|
| | **2004** | **2010** | |
| **NATIONAL** | 0.3813 | 0.4088 | 7.21 |
| **STATES** | | | |
| Abia | 0.3524 | 0.3968 | 12.6 |
| Adamawa | 0.4414 | 0.4339 | −1.7 |
| Akwa ibom | 0.3645 | 0.4381 | 20.2 |
| Anambra | 0.3534 | 0.3803 | 7.6 |
| Bauchi | 0.4705 | 0.3348 | −28.9 |
| Bayelsa | 0.3333 | 0.337 | 1.1 |
| Benue | 0.3888 | 0.4069 | 4.6 |
| Borno | 0.3601 | 0.3841 | 6.7 |
| Cross-river | 0.3977 | 0.4369 | 9.8 |
| Delta | 0.3582 | 0.4698 | 31.1 |
| Ebonyi | 0.3598 | 0.425 | 18.1 |
| Edo | 0.3742 | 0.4177 | 11.6 |

**Table A1.** *Cont.*

| Entities | Income Inequality | | % Change from 2004–2010 |
|---|---|---|---|
| | **2004** | **2010** | |
| Ekiti | 0.3695 | 0.4831 | 30.7 |
| Enugu | 0.3976 | 0.4273 | 7.5 |
| Gombe | 0.3652 | 0.4217 | 15.5 |
| Imo | 0.3844 | 0.425 | 10.6 |
| Jigawa | 0.3368 | 0.3976 | 18.1 |
| Kaduna | 0.3668 | 0.4005 | 9.2 |
| Kano | 0.375 | 0.4692 | 25.1 |
| Katsina | 0.4174 | 0.374 | −10.4 |
| Kebbi | 0.3046 | 0.3259 | 7 |
| Kogi | 0.4914 | 0.4145 | −15.7 |
| Kwara | 0.4848 | 0.3594 | −25.9 |
| Lagos | 0.504 | 0.3719 | −26.2 |
| Nassarawa | 0.3494 | 0.34 | −2.7 |
| Niger | 0.3665 | 0.3675 | 0.3 |
| Ogun | 0.3984 | 0.4076 | 2.3 |
| Ondo | 0.3274 | 0.3869 | 18.2 |
| Osun | 0.3482 | 0.3856 | 10.7 |
| Oyo | 0.3295 | 0.3923 | 19.1 |
| Plateau | 0.4242 | 0.3995 | −5.8 |
| Rivers | 0.4052 | 0.4614 | 13.9 |
| Sokoto | 0.3574 | 0.355 | −0.7 |
| Taraba | 0.3664 | 0.5241 | 43 |
| Yobe | 0.3283 | 0.523 | 59.3 |
| Zamfara | 0.3506 | 0.3397 | −3.1 |
| Federal Capital Territory | 0.4062 | 0.5116 | 26 |

Source: NBS Nigeria Poverty Profile 2010.

**Table A2.** 2004/2010 poverty numbers for absolute, relative and dollar/day poverty.

| Regions & Entities | | Absolute Poverty | | Poor Below 2/3 of the Weighted Mean Household Per Capita Expenditure Regionally Deflated (Relative Poverty) | | Dollar Per Day Based on an Adjusted PPP | |
|---|---|---|---|---|---|---|---|
| | | **Revised 2010** | **2004** | **2010** | **2004** | **2010** | **2004** |
| Sector | Urban | 51.20 | 52.20 | 61.8 | 43.2 | 52.4 | 40.1 |
| | Rural | 69.00 | 73.40 | 73.2 | 63.3 | 66.3 | 60.6 |
| | National | 63.46 | 64.50 | 69.37 | 54.1 | 61.64 | 51.69 |
| Zone | North Central | | | 67.5 | 67 | 59.7 | 58.6 |
| | North East | | | 76.3 | 72.2 | 69.1 | 64.8 |
| | North West | | | 77.7 | 71.2 | 70.4 | 61.2 |
| | South East | | | 67 | 26.7 | 59.2 | 31.2 |
| | South South | | | 63.8 | 35.1 | 56.1 | 47.6 |
| | South West | | | 59.1 | 43 | 50.1 | 40.2 |
| States | Abia | 50.20 | 40.90 | 63.4 | 22.27 | 57.8 | 28.01 |
| | Adamawa | 77.80 | 76.60 | 80.7 | 71.73 | 74.3 | 68.91 |
| | Akwa Ibom | 51.00 | 56.80 | 62.8 | 34.82 | 53.8 | 46.04 |
| | Anambra | 53.70 | 41.40 | 68 | 20.11 | 57.4 | 30.39 |
| | Bauchi | 84.00 | 87.80 | 83.7 | 86.29 | 73.1 | 76.51 |
| | Bayelsa | 44.00 | 40.00 | 57.9 | 19.98 | 47 | 26.29 |
| | Benue | 73.60 | 64.70 | 74.1 | 55.33 | 67.2 | 42.84 |
| | Borno | 60.60 | 59.80 | 61.1 | 53.63 | 55.1 | 48.65 |
| | Cross-Rivers | 60.40 | 67.00 | 59.7 | 41.61 | 52.9 | 51.64 |
| | Delta | 53.80 | 70.60 | 70.1 | 45.35 | 63.6 | 62.28 |
| | Ebonyi | 82.90 | 63.20 | 80.4 | 43.33 | 73.6 | 46.06 |
| | Edo | 64.10 | 53.60 | 72.5 | 33.09 | 66 | 44.31 |
| | Ekiti | 55.90 | 60.40 | 59.1 | 42.27 | 52.6 | 35.51 |
| | Enugu | 60.60 | 50.20 | 72.1 | 31.12 | 63.4 | 33.89 |
| | Gombe | 81.60 | 73.10 | 79.8 | 77.01 | 74.2 | 66.34 |
| | Imo | 39.40 | 46.70 | 57.3 | 27.39 | 50.7 | 26.46 |
| | Jigawa | 88.50 | 95.30 | 79 | 95.07 | 74.2 | 89.54 |

**Table A2.** *Cont.*

| Regions & Entities | | Absolute Poverty | | Poor Below 2/3 of the Weighted Mean Household Per Capita Expenditure Regionally Deflated (Relative Poverty) | | Dollar Per Day Based on an Adjusted PPP | |
|---|---|---|---|---|---|---|---|
| | | Revised 2010 | 2004 | 2010 | 2004 | 2010 | 2004 |
| States | Kaduna | 64.00 | 54.20 | 73 | 50.24 | 61.8 | 37.72 |
| | Kano | 70.40 | 59.40 | 72.3 | 61.29 | 66 | 46.7 |
| | Katsina | 77.60 | 72.90 | 82 | 71.06 | 74.8 | 60.42 |
| | Kebbi | 72.50 | 90.80 | 80.5 | 89.65 | 72.5 | 86.2 |
| | Kogi | 67.40 | 91.80 | 73.5 | 88.55 | 67.3 | 87.46 |
| | Kwara | 72.10 | 87.80 | 74.3 | 85.22 | 62 | 79.85 |
| | Lagos | 40.30 | 69.40 | 59.2 | 63.58 | 49.3 | 64.05 |
| | Nassarawa | 78.40 | 66.10 | 71.7 | 61.59 | 60.4 | 48.17 |
| | Niger | 51.00 | 64.40 | 43.6 | 63.9 | 33.9 | 56.01 |
| | Ogun | 57.60 | 49.90 | 69 | 31.73 | 62.5 | 29.84 |
| | Ondo | 57.70 | 62.80 | 57 | 42.14 | 46.1 | 41.47 |
| | Osun | 37.50 | 44.60 | 47.5 | 32.35 | 38.1 | 22.66 |
| | Oyo | 50.80 | 38.00 | 60.7 | 24.08 | 51.8 | 19.28 |
| | Plateau | 72.40 | 68.50 | 79.7 | 60.37 | 74.7 | 46.78 |
| | Rivers | 47.20 | 56.70 | 58.6 | 29.09 | 50.6 | 43.12 |
| | Sokoto | 86.10 | 75.20 | 86.4 | 76.81 | 81.9 | 70.54 |
| | Taraba | 68.30 | 60.50 | 76.3 | 62.15 | 68.9 | 54.07 |
| | Yobe | 81.70 | 88.00 | 79.6 | 83.25 | 74.1 | 74.12 |
| | Zamfara | 67.50 | 84.00 | 80.2 | 80.93 | 71.3 | 73.38 |
| | FCT | 45.50 | 53.30 | 59.9 | 43.32 | 55.6 | 46.98 |

Source: NBS Nigeria Poverty Profile 2010.

**Table A3.** Nigeria real GDP profile, 2004/2010.

| Year | Real GDP (N Billion) at 1990 Constant Prices |
|---|---|
| 2004 | 527.6 |
| 2010 | 775.5 |

Source: NBS GDP Report for Various Years.

**Appendix B.**

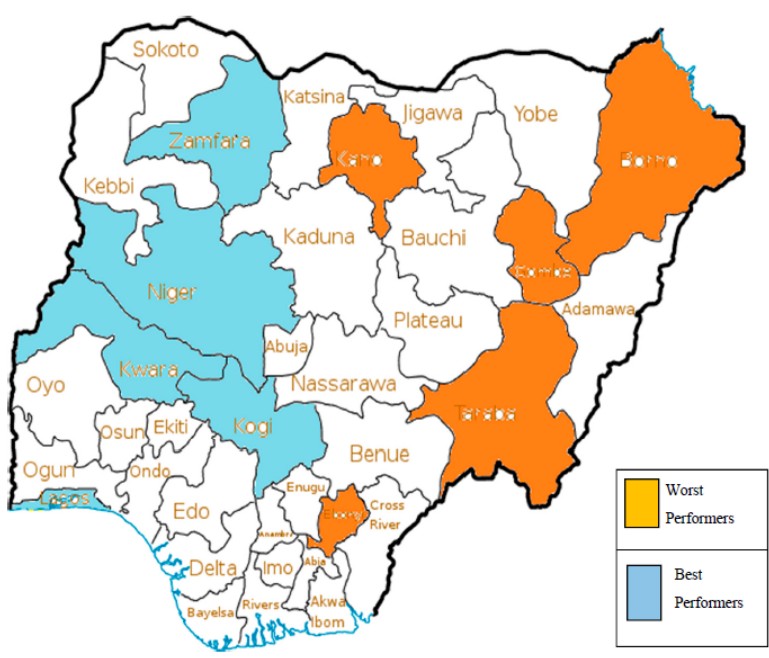

**Figure A1.** Five best and worst performers in poverty reduction using Absolute Poverty Measure.

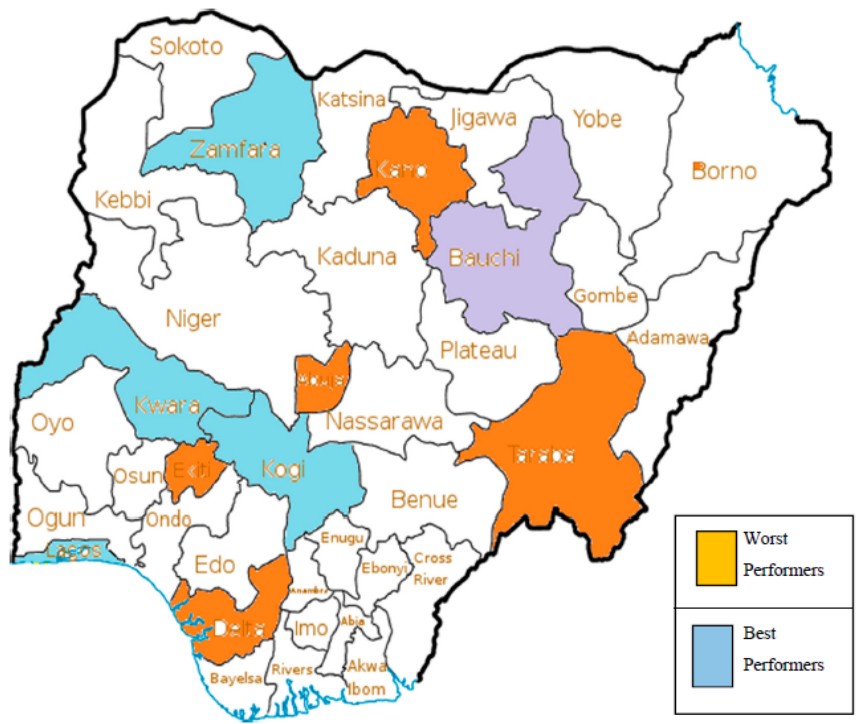

**Figure A2.** Five best and worst performers in poverty reduction using Relative Poverty Measure.

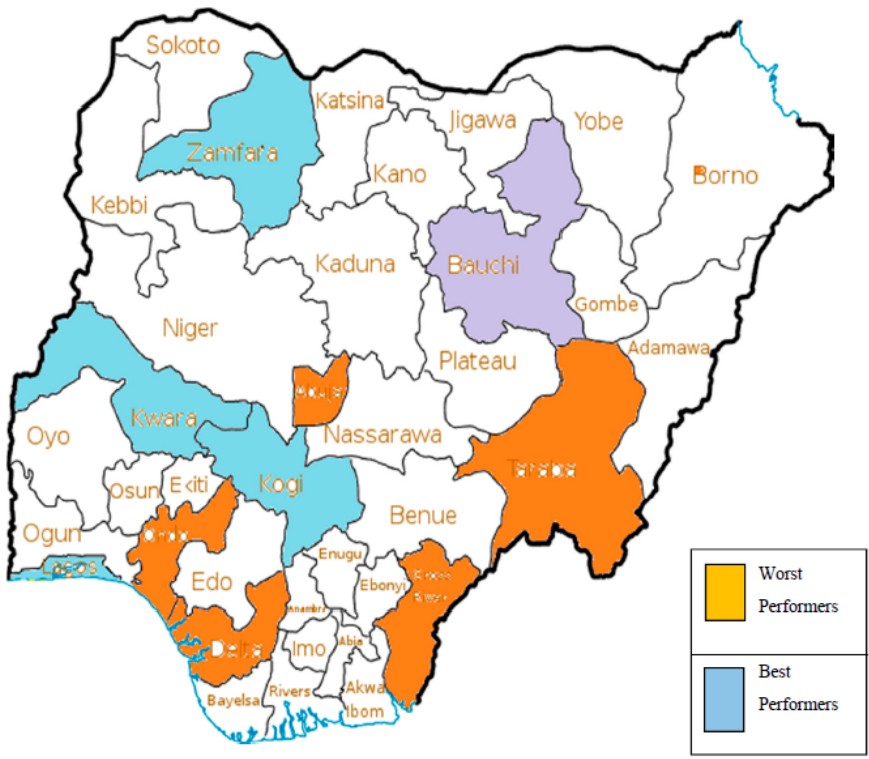

**Figure A3.** Five best and worst performers in poverty reduction using Dollar/day Poverty Measure

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
