# Peer review of "An Investigation of (Non-) Inclusive Growth in Nigeria’s Sub-Nationals: Evidence from Elasticity Approach"

_economies, doi:10.3390/economies5040043_

Round 1
Reviewer 1 Report
1. Rewrite the Abstract -- more to the point. Clarify; don't dazzle.
2. More descriptive title. Maybe: "The implications of inclusive and non-inclusive growth for poverty and inequality in Nigeria's sub-nationals (if that is a word): Evidence from the elasticity approach." Something like that which tells the readers what the paper is all about.
3. Introduction - edit. I would like to see (a) the the objective of the paper clearly stated; (b) why it is important to pursue it -- contribution to existing knowledge; (c) methods + data = methodology statements; (d) results and implications of the results for policy and or future research. You have all these elements already; just make them succinct.
4. The lit review is very good. However, it would be even better if you were able to link more tightly than you have now the paragraph on the gap in literature with the paper research problem. What value does your effort add to what is already known?
5. I like the Methodology, but it would be easier to follow if you could find a way to connect your key assumptions and the Scenarios you put forward on Page 6 of 19. The scenarios in Table 2 are very interesting. You say you compiled them yourself, how, from what? There are standard derivations for these, or are they intuitive? I am not asking you to withdraw them, just to clarify!
6. The results are well presented, but not well discussed -- too many unnecessary details. Instead of explaining columns in Table 3 themselves, for example, show the readers how that is related to the variable interactions using the measures of poverty you have on pp. 8-10. By the way, I could not see your description of the inequality variable; I know what it is, but only because I am an economist. Not all your readers are economists.
7. My suggestion in #6 above applies to Table 4. The table is good; just give the readers a clear way of comprehending its contents
8. I like the "test of hypothesis" in Table 5. Briefly tie these findings to the information you have in your performance maps in the Appendix. Why are some states doing better than others? Are there biases -- geographical, ethnic, religious, resource endowments, etc?
Good paper!
Author Response
Let us begin by thanking the reviewer for the kind words. For us it is everything, not minding whether the paper eventually goes through or not.
1. We acknowledge that this paper makes some shifts from the received knowledge on the subject therefore we take every suggestion in good faith. For the abstract we have reread again and again in search for the dazzling words or sentence(s). Everything stated in the abstract is what is in the body of the work. For instance the scenario matrix is very important to the study. The sub-national we studied lack GDP data so we made a key assumption for this and the last sentence talks about the result of the study. All these are aptly justified.
2. The strength of the work is in the analysis (index). However the title has been modified.
3. We tried to make the introductory paragraphs brief enough. Notice that its contribution to existing knowledge is put at section 2.4, stating the gaps that it fills. This is true because at the end, the body of knowledge we propose will be reviewed as some literature in future.
4. It is both intuitive and standard. It uses microeconomic foundations of Pareto optimality.
5. It is important to explain the columns because it’s an index (as defined by the authors).
6. Inequality is discussed in the scenarios section from line 242.
7. It must be stated that the easiest way to understanding table 4 is to grasp the scenario matrix of the different qualities of growth.
8. The paragraph of line 365 already discusses the results in terms of the biases of some states. Eg as in agrarian, commercial etc.
Reviewer 2 Report
Major Revisions
The title of the paper has to be modified. The term 'inclusive growth' does not suit with the content of the article. Rather I suggest 'Pro-Poor'. The term ‘inclusive’ growth is often used interchangeably with a suite of other terms, including ‘broad-based growth’, ‘shared growth’, and ‘pro-poor growth’. However, inclusive growth is more border than the 'pro poor growth. Elasticity approach is a static approach and may not capture the dynamism embedded with the concept of inclusive growth. To analyze inclusive growth, as the author(s) mentioned in the conclusion section it needs further detailed analysis with longitudinal survey data. Pleased read Commission on Growth and Development (2008) Growth Report: Strategies for Sustained Growth and Inclusive Development, the World Bank.
To enrich the analysis in addition to the data between 2004 and 2010; it would be good if there is data after 2010. However, this depends on data availability.
In your test of hypothesis on the basis of Table 5 percentages of non-inclusive growth are higher in all the columns. Please revisit your conclusions.
Line 391- it reads that "...we can confidently reject the null hypothesis...." But how? How do we know these values are statistically significant. The author(s) need to do some sort of T-test to compare the statistical differences of the values under discussion.
Line 419- Section five- better to state it as...Conclusion and Policy Implications, without 5.1 and 5.2. The recommendations are not supported by the data, like the issue of land redistribution. Land redistribution is still controversial strategy in reducing inequality and inducing economic growth. Furthermore, there is no common yardsticks to make recommendations about Nigeria in the context of China, Korea and Japan. The authors also suggest modern capital intensive technology for rural based economies in the country. Is this not contradiction?
Line 430- galloping inflation? What about the re-distributive power of inflation?
Please revisit.
Minor Suggestions
In section 1, introduction, a concise review of the political economy of Nigeria is worth mention in one of two paragraphs. Words like "tailspin" , " chorus " galloping ...and others can be replaced by more common synonyms to avoid confusion and to ensure easy read.
Line 162- It is also good to write Arabic numbers less ten in words. For example, on line 162 you find .... 2 major problems , better to write it as two major...
Line 182- please specify the $/N exchange rate at the time of analysis
Line 240- equation 4 requires explanations
Line 265- Table 3 can be presented in landscape
Line 367- substitute the phrase ...it is interesting ...it sounds exaggerated
Line 440- Data Tables can be condensed by adding multiple rows as in the next table.
Author Response
Major suggestions
· The authors admire the expansive knowledge of the expert reviewer on the semantics of inclusive growth and pro-poor growth. However, we wish to draw attention to the body of methods used in the work. The study proposed what it called a scenario matrix in table 2 of line 254. With what it calls “quality of growth” the definitions of inclusive growth and its delineations are properly captured. We don’t just look between two extremes of either pro-poor or pro-rich, we also took cognisant of a middle ground analysis. With this the study makes some valid departure from the norm in poverty analysis.
Again the elasticity model is not a static model. It builds in a time lag of six years. This is true as the variables are not all in levels. This is what is called dynamic elasticities. See Dynamic models by Ragnar Nymoen (page 9) http://folk.uio.no/rnymoen/ECON3410_v04_dynamic.pdf
· No there is no data after 2010. Sadly, the authors harped on this in line 439.
· Table 5 shows the number of sub-national that recorded non-inclusive growth. For instance in the absolute poverty measure 54% of the States (that is about 20 States out of 37) suffered non-inclusive growth. Our hypothesis was to check if non-inclusive growth is characterised in all the States (quite an insurmountable task as all the States cannot keep more citizens in poverty at the same time). The result show it is not in all the States.
This brings us to the next suggestion of the need to have a t-test (a standardised value) to be able to reject a null. The t-test creates a model of where we would expect a value for a test statistic to fall which uses the t distribution. It simply answers whether the difference in the sample group is representative of the population. This is an inferential statistical test which is unnecessary for a deterministic process. Looking at it another way, it is inappropriate to apply confidence intervals for this analysis. The Nigeria Bureau of Statistics (NBS) which conducted the household surveys have already standardised the data (from the range of random samples) to arrive at the mean values for the different localities according to the survey report. For this, the report claims the results to be a good representation of the population.
My response is supported accordingly by Petty, M. (n.d). Calculating and using confidence intervals for model validation. https://www.sisostds.org/DesktopModules/Bring2mind/DMX/Download.aspx?Command=Core_Download&EntryId=36208&PortalId=0&TabId=105
· Line 431 says modern technology and not modern capital intensive technology which are two different things. Of course adopting modern technology must not be capital intensive. Also the reviewer suggests that the authors use recommendation like land redistribution is not supported by data. We make bold to say that within the six year span within the data, policies like land ownership reforms and conditional cash transfers were done by some States. Therefore we can confidently say that it is implicit in the data, no wonder the NBS survey measured inequality in different dimensions to include education, health, access to land etc.
In consonance with line 430 we also discussed the issue of redistributive power of inflation when we talked about redistribution not being productivity raising in line 80-81. It just happened that we reiterated it differently in the conclusions.
Minor suggestions
· Political economy of Nigeria can be seen in line 51. Line 162 corrected.
· Line 182 shows the annualised amount for the dollar/day measure. This is not decided by the authors but by NBS from which the data was borrowed and all it showed was the naira equivalent.
· Equation 4 as explained in the text simply shows the growth and inequality elasticities which is computed to measure the index.
· Line 367 corrected.
· Line 440 data tables was adjusted by the editor(s) on how they see appropriate.
Round 2
Reviewer 2 Report
I have still doubt about your claim about Elasticity Approach a dynamic approach. Please supply supporting references.
Authors' revision:
In Section 3.3., authors added "Furthermore, it is important to add that our elasticity model in Equations (1) and (2) are not static but dynamic models because they build in a time lag of six years. This is true as the variables are not all in levels; this is what is called dynamic elasticities. (See Nymoen (2004), page 9)."
Reference:
(Nymoen 2004) Nymoen, Ragner. 2004. Dynamic Models: Lecture Notes. Available online: http://folk.uio.no/rnymoen/ECON3410_v04_dynamic.pdf (accessed on 1 November 2017).